# Anti-SOD1 Nanobodies That Stabilize Misfolded SOD1 Proteins Also Promote Neurite Outgrowth in Mutant SOD1 Human Neurons

**DOI:** 10.3390/ijms232416013

**Published:** 2022-12-16

**Authors:** Meenakshi Sundaram Kumar, Megan E. Fowler-Magaw, Daniel Kulick, Sivakumar Boopathy, Del Hayden Gadd, Melissa Rotunno, Catherine Douthwright, Diane Golebiowski, Issa Yusuf, Zuoshang Xu, Robert H. Brown, Miguel Sena-Esteves, Alison L. O’Neil, Daryl A. Bosco

**Affiliations:** 1Department of Neurology, University of Massachusetts Chan Medical School, Worcester, MA 01605, USA; 2Biochemistry and Molecular Biotechnology Program, Morningside Graduate School of Biomedical Sciences, University of Massachusetts Chan Medical School, Worcester, MA 01605, USA; 3Neuroscience Program, Morningside Graduate School of Biomedical Sciences, University of Massachusetts Chan Medical School, Worcester, MA 01605, USA; 4Department of Biology, Neuroscience and Behavior Program, Wesleyan University, Middletown, CT 06459, USA; 5Horae Gene Therapy Center, University of Massachusetts Chan Medical School, Worcester, MA 01605, USA; 6Department of Biochemistry and Molecular Biotechnology, University of Massachusetts Chan Medical School, Worcester, MA 01605, USA; 7Department of Chemistry, Neuroscience and Behavior Program, Wesleyan University, Middletown, CT 06459, USA

**Keywords:** amyotrophic lateral sclerosis (ALS) (Lou Gehrig disease), antibody engineering, neurite outgrowth, protein misfolding, superoxide dismutase (SOD)

## Abstract

ALS-linked mutations induce aberrant conformations within the SOD1 protein that are thought to underlie the pathogenic mechanism of SOD1-mediated ALS. Although clinical trials are underway for gene silencing of *SOD1*, these approaches reduce both wild-type and mutated forms of SOD1. Here, we sought to develop anti-SOD1 nanobodies with selectivity for mutant and misfolded forms of human SOD1 over wild-type SOD1. Characterization of two anti-SOD1 nanobodies revealed that these biologics stabilize mutant SOD1 in vitro. Further, SOD1 expression levels were enhanced and the physiological subcellular localization of mutant SOD1 was restored upon co-expression of anti-SOD1 nanobodies in immortalized cells. In human motor neurons harboring the *SOD1 A4V* mutation, anti-SOD1 nanobody expression promoted neurite outgrowth, demonstrating a protective effect of anti-SOD1 nanobodies in otherwise unhealthy cells. In vitro assays revealed that an anti-SOD1 nanobody exhibited selectivity for human mutant SOD1 over endogenous murine SOD1, thus supporting the preclinical utility of anti-SOD1 nanobodies for testing in animal models of ALS. In sum, the anti-SOD1 nanobodies developed and presented herein represent viable biologics for further preclinical testing in human and mouse models of ALS.

## 1. Introduction

Dominant mutations in the gene encoding Cu/Zn superoxide dismutase 1 (SOD1) account for 20–25% of the inherited forms of amyotrophic lateral sclerosis (ALS) [1]. Normally SOD1 is present in both the cytoplasm and the nucleus [2]. As a key cytosolic anti-oxidizing enzyme, SOD1 catalyzes conversion of harmful superoxide radicals to hydrogen peroxide and oxygen. SOD1 also functions as a transcription factor in the nucleus, where it regulates genome stability and DNA-damage repair to mitigate oxidative stress [3]. 

Multiple lines of evidence indicate that mutations in SOD1 cause ALS primarily through a gain of toxic function. For instance, overexpression of ALS-linked mutant SOD1 in rodent models recapitulates ALS phenotypes, including motor-neuron degeneration and neuroinflammation [4]. The toxicity of ALS-linked SOD1 may arise from mutation-induced structural perturbations in SOD1 resulting in toxic, misfolded conformations [5,6]. Indeed, ALS-linked SOD1 variants adopt a misfolded and thermodynamically destabilized conformation that correlates with the severity of human disease [7]. Misfolding of SOD1 is also initiated by aberrant post-translational modifications, which are relevant in cases of ALS without SOD1 mutations [5,8,9,10,11]. 

To overcome the toxicity of ALS-linked SOD1, gene-silencing of *SOD1* via microRNAs [12] and antisense oligonucleotides (ASOs) [13,14] has been proposed. However, there may be adverse consequences with these therapeutics as they also reduce expression of wild-type (WT) SOD1 [5,15], given that knockout of SOD1 in mice results in defective axonal homeostasis and stress response, as well as oxidative damage within the nucleus [16,17,18,19]. Misfolded and mutant SOD1 variants also accumulate in the cytoplasm [11,20,21,22,23], which could lead to a loss of nuclear SOD1 activity in a manner that also contributes to ALS and other neurodegenerative disorders [24,25].

Immunotherapy that selectively targets misfolded SOD1 species may represent an alternative and viable therapeutic approach while not reducing expression of SOD1 WT. Active immunization using misfolded SOD1 antigens or passive immunization with antibodies specific to misfolded SOD1 improved survival in transgenic rodent models expressing ALS-linked SOD1 mutations [26,27,28,29,30]. However, the use of conventional monoclonal antibody-based therapy is limited by insufficient pharmacokinetics, high production costs, and a general inefficacy of antibodies for entering cells [31]. Engineered antibody fragments overcome some of these shortcomings. In ALS mouse models, beneficial effects have been observed using anti-SOD1 single-chain variable fragments (scFvs) that consist of only the variable domains of the immunoglobulin-heavy and -light chains [32,33]. Recently, a smaller and more versatile antibody format comprised of a single antigen-binding domain has emerged. These so-called “nanobodies” are derived from the variable domain of heavy-chain-alone antibodies found in camelid sera [34]. With a molecular weight of ~15 kDa, nanobodies are produced efficiently and in high yields as recombinant proteins [34]. Nanobodies are also more effectively delivered into cells through gene-therapy approaches as compared to conventional antibodies [31]. Nanobodies can be further engineered to enhance their performance, such as engaging with cellular protein-degradation machinery [35] and bypassing the blood–brain barrier [36]. In the context of neurological disorders, nanobodies have already shown promise in preclinical models of Parkinson’s [35] and Alzheimer’s disease [37]. However, reports of nanobodies targeting SOD1 remain limited. 

Herein, we developed nanobodies derived from llama sera that exhibit selective reactivity for misfolded SOD1 proteins compared to SOD1 WT. Anti-SOD1 nanobodies did not reduce expression levels of misfolded SOD1 protein in mammalian cells, but rather appeared to stabilize and potentially mitigate the misfolded conformation of mutant SOD1 in cells and in vitro. Co-expression of anti-SOD1 nanobodies led to increased levels of mutant SOD1 in mammalian cells, consistent with enhanced SOD1 stability. Furthermore, co-expression of anti-SOD1 nanobodies increased the nuclear-to-cytoplasmic (N/C) ratio of mutant SOD1 to that of SOD1 WT, suggesting that mutant SOD1 adopts a more wild-type-like conformation when in complex with the nanobodies. Importantly, expression of anti-SOD1 nanobodies exerted a beneficial effect on the health of neurons derived from ALS human induced pluripotent stem cells (iPSCs). These data demonstrate that anti-SOD1 nanobodies have therapeutic potential for modifying the pathogenic properties of mutant SOD1 proteins in vivo. 

## 2. Results

### 2.1. Nanobodies with Selectivity for ALS-Linked SOD1

We sought to identify anti-SOD1 nanobodies that exhibit selectivity for mutant and misfolded forms of SOD1, as such biologics have therapeutic potential. Llamas were immunized simultaneously with recombinant SOD1 WT, SOD1 G93A, and an oxidized form of SOD1 (SODox) that we and others have shown to adopt a mutant-like, misfolded conformation [10,38]. Blood from the immunized animals was used for single-domain antibody, or nanobody, library construction and binder discovery to identify clones that bind SODox over SOD1 WT. Two nanobodies with 96.7% sequence identity, referred to as Nb54 and Nb61, were found to exhibit > 5-fold selectivity for SODox over the native SOD1 WT protein in this screen (Figure 1A). We then expressed and purified Nb54 and Nb61 as recombinant proteins from *E. coli* for further validation and characterization. 

We first assessed the reactivity of Nb54 and Nb61 for recombinant SOD1 proteins by an enzyme-linked immunosorbent assay (ELISA). In addition to the SOD1 variants used as immunogens to create these anti-SOD1 nanobodies (e.g., SOD1 WT, SOD1 G93A, and SODox), we assessed reactivity to SOD1 A4V, representing the most common and aggressive variant in the North American ALS patient population [39]. Relative to SOD1 WT, both Nb54 and Nb61 exhibited 3–4-fold higher reactivity toward SOD1 A4V and SOD1 G93A when tested with 0.12–1 μg/mL concentrations of the respective Nb (Figure 1B). Nb61 also reacted with the denatured form of both SOD1 WT and G93A (Appendix A). Both Nb54 and Nb61 tended to exhibit higher reactivity toward SODox compared to SOD1 WT; however, this difference in reactivity did not reach statistical significance. Given that SOD1 A4V was not used as an immunogen for the generation of these nanobodies, the high reactivity of Nb54 and Nb61 for SOD1 A4V reinforces the notion that ALS-linked SOD1 variants share a common misfolded conformation [6,40]. 

### 2.2. Anti-SOD1 Nanobodies Lead to Enhanced, Rather Than Reduced, Levels of Ectopic SOD1 in Cellulo

Intracellular clearance of nanobody-bound antigen generally does not occur in the absence of a proteolytic targeting signal, such as the PEST degron. PEST sequences are rich in proline, glutamate, serine, and threonine residues, and are found in proteins with particularly short half-lives [41]. Fusion of PEST sequences to proteins, including antibody fragments that recognize neurogenerative-disease-associated proteins a-synuclein [35,42,43] and Huntingtin [44], induce their degradation through the ubiquitin proteasome pathway. Here, we engineered versions of both Nb54 and Nb61 with a C-terminal PEST sequence. We initiated characterization of the SOD1/Nb interaction in cellulo using HEK293T cells, as this represents a tractable cell line with high transfection efficiency. To test whether the nanobodies could induce degradation of ALS-linked SOD1 variants, HEK293T cells were co-transfected with either myc-tagged SOD1 WT (Figure 2A), SOD1 A4V (Figure 2B), or SOD1 G93A (Figure 2C), together with either nanobody, nanobody-PEST, or an empty control plasmid (i.e., the nanobody plasmid without the nanobody gene; Figure 2A–I). Cells were stained 24 h post transfection with anti-nanobody and anti-myc antibodies for detection of nanobody and ectopic SOD1-myc (WT, A4V, and G93A), respectively (Figure 2A–C). As the fluorescent signal intensity is indicative of protein abundance, we measured the mean gray values or fluorescence intensities corresponding to SOD1-myc and nanobody on a per-cell basis. Co-transfection of either Nb54 or Nb54-PEST resulted in significantly higher SOD1-myc signal intensities for all SOD1 variants relative to control cells that were co-transfected with that SOD1 variant and the empty vector (Figure 2D,F,H). Similar results were obtained from cells co-transfected with SOD1 WT and Nb61 or Nb61-PEST (Figure 2E). SOD1 A4V-expressing cells co-transfected with Nb61 or Nb61-PEST showed no difference in SOD1-myc signal intensity compared to cells co-expressing SOD1 A4V and the empty vector (Figure 2G). Although SOD1 G93A-expressing cells co-transfected with Nb61 or Nb61-PEST demonstrated a significant increase in SOD1 G93A signal intensity, the effect was more robust with Nb54 (Figure 2I). 

We also examined the SOD1-myc levels by Western blot analysis of the cell lysates from the HEK293T co-transfection experiments (Appendix A). In contrast to the per-cell fluorescence-intensity analyses (Figure 2), the outcomes of the lysate-based Western blot analyses were variable among experiments (Appendix A), likely due to the variation of transgene expression across a population of cells that have undergone transient co-transfection. To examine this further, we performed a linear-regression analysis of fluorescence intensity corresponding to anti-myc versus anti-Nb for cells co-expressing either myc-tagged SOD1 WT (Figure 3A), SOD1 A4V (Figure 3B), or SOD1 G93A (Figure 3C) with the various nanobody constructs. For all SOD1-myc and nanobody comparisons, including nanobody-PEST constructs, there was a positive correlation between SOD1-myc fluorescence intensity and nanobody fluorescence intensity on a per-cell basis (Figure 3), indicating that the nanobodies generally enhance SOD1-myc expression.

In sum, the outcomes of the fluorescence-intensity analyses are consistent with an association between SOD1-myc and nanobody proteins in cellulo. Unexpectedly, the PEST sequence was ineffective at targeting mutant SOD1 to the proteasome, as SOD1 signal intensities generally increased (rather than decreased) upon co-expression with nanobody–PEST constructs (Figure 2 and Figure 3). These observations suggest that the PEST sequences are not exposed or functional when fused to these anti-SOD1 nanobodies.

### 2.3. The Subcellular Localization of Mutant SOD1 Is Restored by Co-Expression of Anti-SOD1 Nanobodies

The fluorescence-intensity analyses also revealed myc-tagged SOD1 WT expression in both the nucleus and cytoplasm, consistent with previous reports of subcellular SOD1 localization in mammalian cells and nervous tissue [2,45]. Conversely, SOD1 A4V and SOD1 G93A expression were more cytoplasmic relative to SOD1 WT (Figure 4A), possibly due to a misfolded conformation that favors cytoplasmic SOD1 localization [11,23]. To quantify this phenotype, integrated fluorescence-signal intensities of SOD1-myc were measured in the nucleus (defined by DAPI) and the cytoplasm (defined by phalloidin), and nuclear to cytoplasmic ratios (N/C) were determined for each condition. N/C measurements were significantly lower for SOD1 A4V and G93A compared to SOD1 WT, which exhibited a mean N/C of ~1, indicating similar levels of SOD1 WT in the nucleus and cytoplasm (Figure 4B). Although co-transfection of Nb54 or Nb54-PEST did not affect the N/C of SOD1 WT (Figure 4C), these nanobodies resulted in a significant increase in the N/C to a mean of ~1 for SOD1 A4V (Figure 4C,D). The N/C for SOD1 G93A likewise increased to a mean of ~1 with co-expression of Nb54, although not with Nb54-PEST (Figure 4E). Similarly, co-expression of Nb61 and Nb61-PEST did not affect the N/C of SOD1 WT (Figure 4F), but these nanobodies significantly increased the N/C for both SOD1 A4V (Figure 4G) and SOD1 G93A (Figure 4H).

### 2.4. Anti-SOD1 Nanobodies Stabilize Mutant SOD1 in Cellulo

Collectively, the co-transfection studies in HEK293T cells are consistent with an association between mutant SOD1 and our anti-SOD1 nanobodies in cellulo. These results also imply that nanobody binding induces a conformational change within mutant SOD1 that favors a WT-like localization in cells. As ALS-linked SOD1 variants adopt a misfolded and thermodynamically destabilized conformation [5,6,7,40,46,47], we investigated the effects of nanobody binding on mutant SOD1 thermal stability. To this end, we employed differential scanning fluorimetry (DSF), a technique that we and others have used to study the stability of recombinant misfolded proteins [48], including ALS-linked SOD1 [49]. With this thermal-shift assay, protein unfolding is monitored using the SYPRO orange dye, which reports on the exposure of hydrophobic regions [50,51]. Thermal denaturation of SOD1 A4V resulted in two melting transitions with a melting temperature (T_m_) of 48.9 °C and 60 °C, respectively (Figure 5), consistent with two differentially metalated SOD1 A4V species with distinct melting temperatures [46]. A T_m_ of 43.8 °C and 48.6 °C was determined for Nb54 (Figure 5A) and Nb61 (Figure 5B), respectively. Co-incubation of SOD1 A4V and Nb54 resulted in a DSF curve that was substantially shifted to the right. Furthermore, there was a new melting transition with a T_m_ of 71.1 °C, indicative of an SOD1 A4V/Nb54 complex with enhanced stability relative to either protein alone. The DSF curve for the SOD1 A4V/Nb54 complex contained another melting transition with a T_m_ 45 °C, which likely represents unbound SOD1 A4V and/or unbound Nb54 (Figure 5A). The DSF curve for the SOD1 A4V/Nb61 complex was also shifted toward higher temperatures (Figure 5B). Although the individual melting transitions were not well resolved in the case of SOD1 A4V/Nb61, the DSF curve also contained a peak at a high temperature (~85 °C), consistent with a melting transition for SOD1 A4V/Nb61 with enhanced thermostability. In sum, these data indicate that complex formation between anti-SOD1 nanobodies and SOD1 A4V have a stabilizing effect on the mutant protein. A DSF analysis for SOD1 WT was not pursued due to the high thermostability (T_m_ > 90 °C) of this protein, also reported by others [46,49]. 

### 2.5. Anti-SOD1 Nanobody Expression Is Non-Toxic and Induces Enhanced Neurite Outgrowth in Human SOD1 A4V Motor Neurons

Aiming to study our nanobodies in a disease-relevant context without SOD1 overexpression, we generated lentiviral particles for transduction and expression of Nb61, Nb61-PEST, or GFP in human iPSC-derived motor neurons. Virus expressing GFP served as a negative control, as all of the lentiviral constructs were designed to co-express GFP from an internal ribosome entry site (IRES) for identification of transduced cells. Lentiviral particles were delivered to human SOD1 A4V neurons (Figure 6A), which reportedly exhibit reduced cell health compared to control lines without SOD1 mutations [52,53]. Lentiviral transduction efficiencies were typically similar (40–60%) across conditions (Figure 6B). Seven days post viral transduction, we assessed neuronal health by comparing total neurite lengths (anti-NFH) between constructs (Figure 6C) [53]. Compared to SOD1 A4V neurons expressing the GFP control lentivirus, SOD1 A4V neurons expressing Nb61 and Nb61-PEST exhibited a greater total neurite length, which reached statistical significance with Nb61-PEST (Figure 6D). SOD1 A4V neurons were also stained with anti-SOD1, allowing for quantification of endogenous SOD1 fluorescence intensity within transduced GFP-positive cells (Figure 6E). Expression of either Nb61 or Nb61-PEST resulted in higher SOD1 signal intensities in SOD1 A4V neurons (Figure 6F). These results are consistent with the enhanced SOD1-myc fluorescence intensities observed under most conditions in HEK293T cells upon co-expression of anti-SOD1 nanobodies (Figure 2). Whether there are differences in N/C between conditions is unknown, as occurrences of cell clumping and the presence of non-neuronal cells (that also express SOD1) in iPSC-derived neuronal cultures precluded a rigorous N/C analysis, as we showed for HEK293T cells.

We also transduced SOD1 WT neurons with our set of lentiviral constructs, which were generated and handled using the same procedures and reagents as for the SOD1 A4V neurons above [54]. Consistent with other reports [52,53], we routinely observed that the viability and neurite outgrowth of SOD1 WT neurons is greater than SOD1 A4V neurons, which is reflected by the differences in cell and neurite densities of the respective neuronal lines upon thawing (compare Figure 6A and Figure 7A). As for SOD1 A4V lines, lentiviral transduction efficiency of SOD1 WT neurons was similar across all conditions (Figure 7B). Compared to the GFP control, neurite length was not significantly affected by expression of Nb61 or Nb61-PEST (Figure 7C,D). Therefore, expression of anti-SOD1 Nb61 was not toxic to human neurons under these conditions, but instead conferred a health benefit to the SOD1 A4V line. 

### 2.6. Anti-SOD1 Nanobodies Detect Human SOD1 G93A in Lysates from an ALS Mouse Model

SOD1-G93A transgenic rodent models are used most for preclinical testing of SOD1-based therapeutics in the ALS field ([4,14]). To investigate the preclinical therapeutic utility of our anti-SOD1 nanobodies, we tested whether Nb54 could detect ectopic human SOD1 G93A in lysates prepared from *SOD1^G93A^* mouse spinal cord tissue with a competitive ELISA as follows. Recombinant SOD1 G93A was coated onto wells of the ELISA plate, and the binding of Nb54 to the immobilized SOD1 G93A was measured as a function of increasing amounts of a “competing” antigen (Figure 8A). In the absence of *SOD1^G93A^* lysate or an otherwise Nb54-reactive antigen, maximal binding of Nb54 to the immobilized SOD1 G93A was expected (Figure 8A; top). In the presence of a competing antigen that binds Nb54, there was less available Nb54 to react with the immobilized SOD1 G93A in the ELISA plate, and thus reduced signal in the assay (Figure 8A; bottom). As Nb54 binds to recombinant SOD1 G93A, increasing concentrations of this antigen was used as a positive control to verify competition with Nb54 and a reduced signal in the ELISA (Figure 8B). In contrast, spinal cord lysates derived from non-transgenic (Non-Tg) animals were unable to compete in this assay, even at the highest concentration tested, indicating a lack of reactivity between Nb54 and endogenous murine SOD1 WT. However, the same Non-Tg lysates spiked with recombinant SOD1 G93A did compete with Nb54/immobilized SOD1 G93A binding. Importantly, *SOD1^G93A^* spinal cord lysates effectively competed with Nb54/immobilized SOD1 G93A binding in a dose-dependent manner, indicating that Nb54 reacts with ectopic human SOD1 G93A in this lysate (Figure 8B). These results demonstrate target engagement between Nb54 and human SOD1 G93A in a complex biological mixture and indicate that there is no cross-reactivity between Nb54 and murine SOD1 WT, thus providing a foundation for future preclinical investigation of Nb54 in *SOD1^G93A^* mice.

## 3. Discussion

In this study, we developed and characterized two anti-SOD1 nanobodies, Nb54 and Nb61, as potential therapeutic molecules for ALS. Nanobodies can be engineered to direct their cognate antigens to different cellular machineries, thereby serving as versatile tools for managing intracellular SOD1 [35]. Contrary to our initial hypothesis, the addition of a PEST tag to Nb54 or Nb61 did not result in reduced SOD1 levels. This may be due to changes in structural properties of the PEST tag, such as poor solvent accessibility, upon fusion with Nb54 and Nb61. Different outcomes may be achieved by engineering a spacer sequence between the nanobody and PEST sequences and/or placing the PEST sequence at the N-terminus (as opposed to the C-terminus herein). Irrespective of the presence of PEST tag, both Nb54 and Nb61 enhanced SOD1-myc signal intensities in HEK293T cells. In contrast, no changes in SOD1 levels were observed with anti-SOD1 nanobodies from a different source [55]. In the case of Nb54-PEST, signals for myc-tagged SOD1 WT and G93A were higher in the co-transfection studies compared to nanobody without PEST. Similarly, the effects of Nb61-PEST were more pronounced than Nb61 in human SOD1 A4V neurons. Therefore, the PEST sequence may also induce structural changes within the nanobodies that in turn impact the Nb/SOD1 interaction. These results highlight the potential for further optimization of these anti-SOD1 nanobodies for SOD1 target engagement. 

Anti-SOD1 nanobodies also affected the nucleocytoplasmic distribution of mutant SOD1 proteins in HEK293T cells. Unlike SOD1 WT, which is expressed in both the nucleus and cytoplasm, ALS-linked SOD1 mutants exhibit enhanced cytoplasmic localization that was observed here and reported previously by others [11,23,56,57]. Cytoplasmic localization of mutant SOD1 is likely a result of mutation-induced misfolding, which could expose a putative nuclear export signal and thus nuclear export of mutant SOD1 via CRM1 (Chromosomal Maintenance 1) [23]. Co-expression of our anti-SOD1 nanobodies restored mutant SOD1 in the nucleus to SOD1 WT levels. This appears to be a unique property of our nanobodies that was not reported for other anti-SOD1 intrabodies [32,33,55]. In this regard, the activity of our nanobodies may resemble the macrophage migration inhibitory factor, a chaperone-like protein that also restores the N/C of mutant SOD1 [56]. It is unlikely that Nb54 or Nb61 sequesters SOD1 within the nucleus, as both nanobodies are expressed throughout the nucleus and cytoplasm (Figure 2). Rather, we speculate that binding of Nb54 or Nb61 to mutant SOD1 converts misfolded SOD1 into a more SOD1 WT-like conformation, thereby favoring the nuclear localization observed for SOD1 WT. This model is supported by the outcomes of our DSF studies, which revealed that both Nb61 and Nb54 exert a stabilizing effect when in complex with SOD1 A4V. 

We noted some differences in the properties of our anti-SOD1 nanobodies when assessed in vitro as recombinant proteins by ELISA versus when expressed in cellulo. For example, Nb54 and Nb61 exhibited similar reactivities for SOD1 A4V and G93A in the ELISA, whereas Nb54 exerted a more pronounced effect on SOD1 signal intensities and mutant SOD1 N/C localization in HEK293T cells relative to Nb61. These results suggest that the physicochemical properties of the nanobodies and/or their capacity to interact with SOD1 proteins is influenced by additional factors in cellulo. Further, Nb54 and Nb61 were selective for both SOD1 A4V and SOD1 G93A over SOD1 WT in vitro but appeared to engage with and enhance levels of myc-tagged SOD1 WT when co-expressed in HEK293T cells. One explanation could be that Nb/SOD1 WT interactions are facilitated by the overexpression conditions used for co-transfection studies. We expect that some degree of anti-SOD1 nanobody target engagement with SOD1 WT will not preclude their therapeutic utility, as ectopic Nb61 expression was not toxic to human neurons expressing SOD1 WT or mutant SOD1 A4V. Furthermore, enhancement of SOD1 WT levels may be preferred and possibly beneficial over a reduction in SOD1, particularly in a disease context for which there is elevated oxidative stress [3,5,15,24,58,59]. Additionally, a beneficial effect has been observed in ALS-SOD1 models upon treatment with diacetyl-bis(4-methylthiosemicarbazonato) copper(II) (Cu(II)(atsm)), which promotes metalation and increases the levels of SOD1 [60,61,62]. Thus, anti-SOD1 nanobodies offer an alternative approach to current gene silencing strategies that target both mutant and WT *SOD1* alleles, which causes an overall reduction in SOD1. 

To assess the potential of our anti-SOD1 proteins for preclinical studies and eventual therapeutic application, we performed studies in ALS-relevant models including human iPSC-derived motor neurons harboring the SOD1 A4V mutation. SOD1 A4V is the most common and aggressive ALS-linked mutation in North America, and therefore biologics targeting this protein are expected to have high therapeutic value for the ALS field. Although SOD1 A4V was not used as an antigen for our anti-SOD1 nanobodies, Nb54 and Nb61 exhibited selectivity for this mutant protein. This observation raises the possibility that our anti-SOD1 nanobodies could be reactive for other SOD1 variants, of which there are over 170 reported to date [63]. Transduction of lentiviruses expressing Nb61 and Nb61-PEST were not toxic to human neurons but rather conferred a beneficial effect on the health of SOD1 A4V neurons. Both neurite outgrowth and SOD1 levels were enhanced upon expression of Nb61 compared to the control condition. In addition to human neurons, we also assessed the utility of our anti-SOD1 nanobodies for preclinical studies using the *SOD1^G93A^* mouse model [4]. Our results demonstrate that Nb54 binds human SOD1 G93A in the context of mouse spinal cord lysate. Together, these proof-of-concept studies warrant future investigations with a larger panel of human neurons harboring different ALS-linked SOD1 mutations and a cohort of *SOD1^G93A^* mice to further assess the therapeutic potential of anti-SOD1 nanobodies for ALS. 

In sum, the nanobodies developed and characterized herein appear to stabilize the physiological conformation of SOD1. This stabilization may underlie the restoration of mutant SOD1 to normal subcellular locations in immortalized cells and confer protection in otherwise unhealthy human ALS SOD1 A4V neurons. Given that mutant SOD1 instability appears to correlate with ALS disease severity in humans [7], we propose that boosting levels of functional and natively folded SOD1 with anti-SOD1 nanobodies is a viable therapeutic direction for treating ALS.

## 4. Materials and Methods

### 4.1. Generation of Anti-SOD1 Nanobodies

All recombinant SOD1 proteins used throughout this study, including antigens for the generation of anti-SOD1 nanobodies, were expressed and purified as described previously by our lab [6,38]. SODox was generated as described [38]. To create anti-SOD1 nanobodies, two llamas were immunized with SOD1 WT, familial ALS-linked SOD1 G93A and SOD1ox by Triple J Farms/Kent Laboratories (Bellingham, WA, USA). Blood samples collected from both immunized animals were used to construct a nanobody gene library by GenScript USA Inc. through a single-domain antibody (sdAb) library construction and binder discovery package SC1590. Briefly, total RNA was extracted from llama blood samples using Trizol. Nanobody-encoding genes (V_H_H) were RT-PCR cloned and amplified from the mRNA of peripheral blood mononuclear cells (including B-cells). The library was constructed by transformation of nanobody/V_H_H DNA fragments into phage-display SS320 chemically competent *E. coli* cells. Based on the number of transformants on the agar plates, the library size was estimated at >1.75 × 10^9^ unique sequences/clones. The library was then screened using phage display by the CRO GenScript USA Inc. for clones that exhibit selectivity for SODox and counterscreened to exclude clones that exhibit high reactivity for the native SOD1 WT protein. Two clones that exhibited > 5-fold selectivity for SODox over the native SOD1 WT by enzyme-linked immunosorbent assay (ELISA) were pursued for additional studies. 

### 4.2. Plasmid Construction

Nanobody sequences were sub-cloned into the pTP212 plasmid (a kind gift from Dr. Dirk Gorlich, Max Planck Institute for Biophysical Chemistry, Göttingen, Germany) for recombinant bacterial expression of nanobody protein containing an N-terminal His-SpbrNEDD8 tag [64]. The plasmid encoding the His-MBP-brSUMO-bdNEDP1 enzyme, which cleaves the SpbrNEDD8 tag, was also a kind gift from Dr. Dirk Gorlich [65]. Nanobody plasmids with and without a C-terminal PEST signal sequence (SHGFPPEVEEQDDGTLPMSCAQESGMDRHPAACASA RINV) for transient transfection into mammalian cells were synthesized as DNA G-blocks and then sub-cloned into the pcDNA 3.1 (−)plasmid (ThermoFisher Scientific, Waltham, MA, USA, V87520) using the NEBuilder HiFi assembly kit (New England Biolabs, Ipswich, MA, USA, E5520S). The same nanobody sequences (with and without PEST) were sub-cloned into the low-expression lentivirus vector CShPW2 for lentiviral expression using the NEBuilder HiFi assembly kit (New England Biolabs, Ipswich, MA, USA, E5520S). The CShPW2 plasmid contains a green fluorescent protein (GFP) reporter expressed downstream of an internal ribosome entry site (IRES) and independently of the nanobody sequence [66]. Plasmids for mammalian expression of SOD1-myc under the cytomegalovirus (CMV) promotor were a kind gift from Dr. Zuoshang Xu (University of Massachusetts Chan Medical School, Worcester, MA, USA).

### 4.3. Recombinant Nanobody Expression and Purification

With the exception of the competition ELISA, all in vitro experiments were performed with nanobody protein prepared as follows. Nanobodies with N-terminal His-SpbrNEDD8 tag were expressed in *Escherichia coli* (*E. coli*) BL21 (DE3) pLysS cells (Millipore Sigma, Burlington, MA, USA, 69451-3). Bacterial cultures (1 L) were grown at 37 °C until the optical density (OD) reached 0.6–0.7. Protein expression was induced by adding isopropyl-beta-D-thiogalactoside (IPTG, Goldbio, St. Louis, MO, USA, I2481C25) to a final concentration of 1 mM. The cultures were further grown at 16 °C for 16 h, after which the cells were harvested by centrifugation and stored at −80 °C until the purification could be initiated.

Bacterial pellets were thawed on ice and resuspended in chilled lysis buffer (50 mM Tris/HCl, 500 mM NaCl, 10 mM imidazole, 1 mg/mL lysozyme, 0.3% NP-40, pH 7.4) supplemented with protease-inhibitor cocktail (Millipore Sigma, Burlington, MA, USA, 11873580001). After sonication, the lysates were clarified by centrifugation at 26,000× *g* for 30 min at 4 °C. The clarified lysate was loaded onto a 1 mL HisTrap HP column (Cytiva, Marlborough, MA, USA, 29051021) equilibrated with buffer A (50 mM sodium phosphate, 300 mM NaCl, 45 mM imidazole, pH 7.0) and subsequently washed with buffer A. Bound proteins were eluted with 50 mM sodium phosphate, 300 mM NaCl, and 500 mM imidazole at pH 7.0. Elution fractions containing nanobody were pooled and concentrated using a centrifugal concentrator (Vivaspin 5000 MWCO, Sartorius, Göttingen, Germany, VS0611) as per the manufacturer’s instructions. The concentrated protein was buffer exchanged into phosphate-buffered saline (PBS, pH 7.4) using a Sephadex-25 desalting column (Cytiva, Marlborough, MA, USA, 17085101). To cleave the His-SpbrNEDD8 tag from the nanobodies, His-MBP-brSUMO-brNEDP1 enzyme was expressed and purified similar to the above protocol. Tag cleavage was performed by incubating nanobody proteins with His-MBP-brSUMO-brNEDP1 enzyme at molar ratio of 1:100 (enzyme:nanobody) in PBS containing 0.25 M sucrose, 2 mM MgCl_2_, and 2 mM dithiothreitol (DTT) for 90 min at 4 °C. The mixture was applied to the 1 mL HisTrap HP re-equilibrated with buffer A and the flow-through containing the untagged nanobody was collected. The untagged nanobody was concentrated and buffer exchanged into PBS as described above and stored at −80 °C. For the competitive ELISA, nanobody with a non-cleavable his-tag was purified similarly to the protocol above. 

### 4.4. ELISA

An indirect ELISA was used to assess the selectivity of the nanobodies for recombinant SOD1 proteins as follows. SOD1 (0.1 μg/50 μL in phosphate buffered saline; PBS) was coated onto 96-well medium-binding microplates (Greiner BioOne, Monroe, NC, USA, 655001) overnight at 4 °C. All subsequent steps were performed at ambient temperature. Coated plates were washed with wash buffer (PBS containing 0.05% (*v/v*) Tween-20) and blocked with a 5% (*w/v*) solution of bovine serum albumin (BSA) in PBS for 1 h. After washing, the plates were incubated for 1.5 h with nanobody diluted (0–1 μg/mL) in wash buffer. Plates were then washed three times with wash buffer and incubated with horseradish peroxide-conjugated anti-nanobody (1:2500, GenScript, Piscataway, NJ, USA, U8401BI120) for 1 h. After three washes, 100 μL of 3,3′,5,5′-Tetramethylbenzidine (TMB, SurModics, Eden Prairie, MN, USA, TMBS-0100-01) were added for 10–30 min. The reaction was quenched with 100 μL Liquid Stop solution (Surfmodics, Eden Prairie, MN, USA, LSTP-0100-01). The absorbance or optical density (OD) at 450 nm for each well was measured with a plate reader (Perkin Elmer, Waltham, MA, USA, Victor X5 model). The average of the OD 450 nm for each independent ELISA experiment (*n* = 3) was determined and statistical analysis was performed as described below and in the figure legend. 

A competitive ELISA was used to assess a potential interaction between nanobody and SOD1 derived from an ALS mouse model as follows. Spinal cord tissue was extracted from four P70-78 B6SJL-Tg(SOD1*G93A)1 Gur/J (*SOD1^G93A^*) mice and four P70-75 WT [4], non-transgenic mice. Tissues were lysed separately in ice-cold 25 mM Tris/Cl, pH 7.8 buffer supplemented with protease-inhibitor cocktail (Millipore Sigma, Burlington, MA, USA, 11873580001) using a Dounce homogenizer. Lysates were clarified by centrifugation at 15,600× *g*, 4 °C, and the total protein concentration was determined by the bicinchoninic acid assay (BCA; ThermoFisher, Waltham, MA, USA, 23227) according to the manufacturer’s instructions. All research involving animals for the following post-mortem tissue processing was reviewed and approved by the Institutional Animal Care and Use Committee (IACUC) at the University of Massachusetts Chan Medical School.

Microplates (96-well) were coated with recombinant SOD1 G93A (0.1 μg/50 μL diluted in PBS) and blocked with BSA as described above. Murine-tissue lysates (25 μL of 3.6 mg/mL) either alone or spiked with recombinant SOD1 G93A (0.2 μg/25 μL) were added to the coated wells. Serial dilutions were prepared in 25 μL of assay buffer (0.2% (*w/v*) BSA in PBS). Recombinant SOD1 G93A (0.2 μg/25 μL) diluted in assay buffer served as a positive control for competition. A total of 25 μL of Nb54 (0.2 μg/mL) diluted in assay buffer was added to all the wells and incubated at ambient temperature for 1 h. Plates were washed and processed as described for the indirect ELISA, except absorbance values were normalized to the signal from Nb54 applied to wells coated with SOD1 G93A in the absence of competing antigen. A statistical analysis was performed as described below and in the figure legend.

### 4.5. Transient Transfection of HEK293T Cells

Human embryonic kidney 293T (HEK293T) cells were maintained in Dulbecco’s modified Eagle’s medium (DMEM; Invitrogen, Waltham, MA, USA, 11965118) containing 10% (*v/v*) fetal bovine serum (MilliporeSigma, Burlington, MA, USA, F4135) and 1% (*w/v*) penicillin-streptomycin (Invitrogen, Waltham, MA, USA, 15140122) at 37 °C and 5% CO_2_. Cells at a density of 1.6 × 10^5^ cells/well were plated in a 24-well plate containing coverslips coated with poly-l-lysine. After 24 h, cells were transiently co-transfected with plasmids encoding nanobody or nanobody-PEST (1 μg) and SOD1-myc, SOD1-A4V-myc, or SOD1-G93A-myc (50 ng) using 3 μL Lipofectamine 2000 (ThermoFisher, Waltham, MA, USA, 11668-019) diluted in OptiMEM (Invitrogen, Waltham, MA, USA, 31985070). Control conditions were included as follows: cells transfected with Nb54 or Nb54-PEST alone, cells transfected with Nb61 or Nb61-PEST alone, and cells co-transfected with an empty vector (the nanobody plasmid without the nanobody gene) and either SOD1-myc, SOD1 A4V-myc, or SOD1-G93A-myc. 

Cells were fixed 24 h post-transfection with 4% paraformaldehyde for 15 min at ambient temperature. For immunofluorescence microscopy, cells were permeabilized with 1% Triton X-100 (Sigma, St. Louis, MO, USA, T9284) for 10 min and blocked with PBSAT (PBS with 1% BSA and 0.5% Triton X-100) for 1 h. Cells were incubated with the rabbit anti-nanobody (1:1000) described above and mouse anti-myc (1:100) for 1 h 15 min. Anti-myc (9E 10) was developed by Bishop, J.M., University of California, San Francisco (UCSF), and was obtained from the Developmental Studies Hybridoma Bank (DSHB), created by the National Institute of Child Health and Human Development (NICHD) of the National Institutes of Health (NIH) and maintained at The University of Iowa (Iowa City, IA, USA) Department of Biology. Cells were then incubated with secondary antibodies anti-rabbit Alexa Fluor 488 (Jackson ImmunoResearch Laboratories, 711-545-152) and anti-mouse Cy3 (Jackson ImmunoResearch Laboratories, 715-165-151) diluted 1:2000 in PBSAT for 1 h. To outline and define the cell boundary, cells were stained with Phalloidin Alexa Fluor 647 (Invitrogen, Waltham, MA, USA, A22287) at 1:100 in PBSAT for 40 min and then counterstained with DAPI (Sigma Aldrich, D9542) for 5 min. Coverslips were mounted using Prolong Gold anti-fade reagent (Cell Signaling Technologies, Danvers, MA, USA, 9071S) with a refractive index of 1.46. 

### 4.6. Image Acquisition and Analysis of Transfected HEK293T Cells

Images were acquired with a Leica DMI 6000 inverted fluorescent microscope equipped with a 40× air lens and a Leica DFC365 FX camera (6.45 μm pixel size) using AF6000 Leica software v.3.1.0 (Leica Microsystems, Wetzlar, Germany). Twelve μm z-stacks (0.44 μm step size, 28 planes) were collected using the Cy5, Cy3, GFP, and DAPI channels (center/band width, nm: excitation 545/39, 620/60, 470/40, 360/40, respectively; emission 605/75, 700/75, 525/45, 470/40, respectively). Stacked images were presented as the maximum-intensity projection of the center five planes. All images were acquired with identical settings within each experiment. Cells that displayed oversaturated signal were excluded from analyses. For co-transfected conditions, only cells that were GFP positive and Cy3 positive were analyzed. The experimentalist was blinded to transfection conditions during the acquisition and data analyses for all the experiments are described below.

#### 4.6.1. HEK293T N/C Ratio

The nuclear and cytoplasmic compartments were defined using DAPI and phalloidin fluorescent signals, respectively. Only cells with distinct and non-overlapping cytoplasmic and nuclear compartments were included in the analysis. The integrated fluorescence intensity of each fluorophore, corresponding to SOD1-myc or nanobody, was measured using a 2 μm × 2 μm square region that was manually placed within the nucleus and cytoplasm of each cell. The square was placed in an area with a signal that was representative of the overall compartment, thus avoiding areas of extreme bright or weak signal [67]. The N/C ratio for each fluorophore was calculated by dividing the intensity of the nuclear signal by the cytoplasmic signal. Cells from three random fields-of-view per condition were analyzed, for a total of 81–200 cells per condition over *n* = 3 independent biological replicates. 

#### 4.6.2. HEK293T Signal Intensity Analysis

The whole-cell outlines described in the colocalization analysis were then used to measure the integrated fluorescence intensity of each cell using FIJI. The mean integrated fluorescence intensity of each fluorophore, corresponding to SOD1-myc or nanobody, was measured for each whole cell and plotted for analysis.

#### 4.6.3. Western Blot Analysis

HEK293T cells transfected as described above in Section 4.5 were washed with PBS and lysed in cold RIPA buffer (Westnet, Canton, MA, USA, BP-115-500) for 20 min on ice, after which the lysates were clarified by centrifugation at 13,000 rpm for 15 min at 4 °C. Protein concentration was estimated by bicinchoninic acid (BCA) assay (Thermo Scientific Pierce, 23227) and 20 μg of total protein was electrophoresed through 15% polyacrylamide gel and transferred onto PVDF membrane (Millipore, Burlington, MA, USA, IPFL00010) for 1 h at 100 V. The immunoblot was blocked for 1 h with blocking buffer (LICOR, Lincoln, NE, USA, 927-70001) and then incubated overnight at 4 °C with the primary antibodies: anti-myc (1:1000, DSHB 9E-10), the anti-nanobody described above (1:2500), anti-tubulin (Sigma Aldrich, St. Louis, MO, USA, T5168, 1:5000), anti-GAPDH (Sigma Aldrich, St. Louis, MO, USA, G8795, 1:2000), and anti-SOD1 (Abcam, Cambridge, UK, ab79390, 1:15,000). Blots were probed for 1 h with IRDye-conjugated secondary antibodies (LICOR, Lincoln, NE, USA) and imaged using the Odyssey Infrared Imager (LICOR, Lincoln, NE, USA, 9120).

For Western blot analysis of recombinant proteins, recombinant SOD1 WT and G93A were denatured by boiling in 1× Laemmli buffer (Westnet, Canton, MA, USA, BP-111R) and subjected to Western blot analysis as described above. The immunoblot was incubated overnight with Nb61 (0.2 μg/mL in PBS) at 4 °C followed by incubation with the rabbit anti-nanobody described above (1:2500) for 2 h at ambient temperature. As a positive control, a duplicate immunoblot was processed with a commercial pan anti-SOD1 antibody (Abcam ab79390, 1:15,000). Blots were probed for 1 h with IRDye-conjugated secondary antibodies (LICOR, Lincoln, NE, USA) and imaged as described above.

### 4.7. iPSC Culture, Motor Neuron Differentiation, and Lentiviral Transduction

Human WT (1016a) and SOD1 A4V ALS-patient (39b) iPSCs were differentiated into motor neurons following previously established 3D methods [54] and were dissociated and cryogenically stored at 21 days of differentiation. Thawed neurons were plated as single cells in 384-well cell-culture plates previously coated with laminin (2.5 μg/mL) and fibronectin (7.5 μg/mL). Cells were plated in complete media, comprised of Neurobasal medium (Gibco, Grand Island, NY, USA, 21103-049), 1× N2 supplement (Gibco, Grand Island, NY, USA, 17502-048), 1× B27 supplement (Gibco, 17504044), 1× Glutamax (Gibco, Grand Island, NY, USA, 35050061), 1× non-essential amino acids (Gibco, Grand Island, NY, USA, 11140-050), 1× penicillin-streptomycin (Gibco, Grand Island, NY, USA, 10378-016), 3.2 mg/mL D-glucose, 20 μM ascorbic acid, 10 ng/mL brain-derived neurotrophic factor (BDNF), 10 ng/mL ciliary neurotrophic factor (CNTF), and 10 ng/mL glial cell-derived neurotrophic factor (GDNF).

Plasmids for lentiviral transduction are described above under “Plasmid Construction.” Lentiviral particles were prepared for GFP alone (negative control), Nb61, and Nb61-PEST at the Viral Vector Core of the Gene Therapy Center within UMass Chan Medical School. Note that GFP is expressed as a reporter and not a fusion protein via these constructs. At the time of plating (8000 cells per well), lentiviral particles were added to the complete media together with 5 μg/mL polybrene for a final viral titer of 10^8^ vp/mL and an estimated MOI of 125. For all experiments, lentivirus expressing GFP alone, Nb61, and Nb61-PEST were tested in parallel within the same plate. At day in vitro (DIV) 7, plates were fixed with 4% paraformaldehyde for 15 min at ambient temperature for staining and immunolabeling as described below. For viral transduction-efficiency calculations, a subset of wells underwent live-cell imaging with Hoechst (nuclear stain, all cells) and propidium iodide (dead cells) at DIV7. Transduction efficiency was calculated as the number of GFP-positive cells divided by the total number of live cells per well across *n* = 3 wells.

### 4.8. Immunofluorescence Microscopy Analysis of iPSC-Derived Motor Neurons

Fixed cells within 384-well plates were blocked (5% FBS, 2% BSA, 0.1% Triton X100 in PBS) and incubated with neurofilament H (NFH) clone SMI-32 (1:1000, Biolegend, San Diego, CA, USA, 801701) and SOD1 (1:500, Enzo Biochem, Farmingdale, NY, USA, ADI-SOD-100-J) primary antibodies overnight, washed, then incubated with animal-matched Alexa-conjugated secondaries and Hoechst counterstain. Plates were imaged on an ImageXpress Pico System (Molecular Devices, San Jose, CA, USA) using automated capture. Image analysis was performed in MetaXpress (Molecular Devices, San Jose, CA, USA, version 6.6.2.46). Technical replicates were defined as an individual well of a 384-well plate. In-plate technical replicates ranged from 4 to 6 wells per condition. Biological replicates were defined as independent experiments from a separate thaw of iPSC-derived motor neurons. Herein, three biological replicates of SOD1 A4V and two of WT motor neurons were analyzed. Additionally, both genotypes of neurons originated from two independent differentiation batches. In the figures, defined symbols denote a specific biological replicate as defined in the figure legend. 

#### 4.8.1. Neurite-Tracing Analysis

Images were acquired at 10× in a Pico high-content imager (Molecular Devices, San Jose, CA, USA) and stitched to create one image containing three fields of view across the well of a 384-well plate. Stitched images were then analyzed with a custom neurite-tracing script written in the MetaXpress image-analysis software (Molecular Devices, San Jose, CA, USA, version 6.6.2.46). In brief, neurite detection was set to be ≥ 3 times the intensity of the background, to be in the width range of 0–5 μm, and to be at least 2 μm long to be counted. All calculated lengths were summed across the stitched fields, covering approximately 50% of the area of a well of a 384-well plate. The analyst was blinded to the conditions.

#### 4.8.2. SOD1-Intensity Analysis

Images were acquired at 10× in a Pico high-content imager (Molecular Devices, San Jose, CA, USA) and stitched as described for neurite tracing. Stitched images were then analyzed with a custom script written in the MetaXpress image-analysis software (Molecular Devices, San Jose, CA, USA, version 6.6.2.46) as follows. GFP-positive cells were identified using the “auto-threshold” tool in the GFP channel. The threshold was set to ≥ 3 times the background intensity. The resulting GFP-positive area mask was then overlaid on the SOD1 channel (i.e., to measure SOD1 intensity only in cells with positive GFP signal, and therefore transduced with the lentiviral constructs). The SOD1 intensity corresponding to the GFP-positive mask was calculated as a total intensity value for each well. The total intensity for each well was then divided by the total area of the GFP+ mask to control for different numbers and different sizes of GFP-positive cells. Cells expressing Nb61 or Nb61-PEST were compared to cells expressing GFP alone (*n* = 3 technical replicates) using GraphPad Prism (v9.3). The analyst was blinded to the conditions. 

### 4.9. Differential Scanning Fluorimetry (DSF)

DSF measurements were performed similarly to our previous report [48]. SOD1 A4V (10 μM, monomeric concentration), Nb61 (10 μM), Nb54 (10 μM), or mixtures of SOD1 A4V and nanobody (10 μM each) were prepared in PBS and incubated on ice for 1 h. SYPRO Orange (Invitrogen, Waltham, MA, USA, #S6651) was then added with a final concentration of 25× with a total reaction volume of 20 μL. All samples were run in duplicate in 384-well plates. The dye diluted in PBS containing no protein served as a negative control. Thermal-scanning and fluorescence measurements were performed with a Bio-Rad C1000 Touch Thermal Cycler with CFX384 Optical Reaction Module (Bio-Rad, Hercules, CA, USA, #1845384). The samples were gradually heated at 0.3 °C/5 sec. Fluorescence measurements were acquired with each temperature increment. The fluorescence intensities from the dye-only reactions were subtracted from the experimental wells. The resulting fluorescence intensities from duplicate experimental wells were averaged and normalized to 1 (i.e., each experimental curve was normalized separately to the highest fluorescence-intensity measurement within that curve) and plotted as a function of temperature. The temperature corresponding to the maxima of the first derivative of each melting transition was used as an estimate of the melting temperature (Tm). All data are representative of at least three independent biological experiments. 

### 4.10. Statistical Analysis

Statistical analysis was performed using Graphpad Prism 9 (v9.3). Kruskal–Wallis with Dunn’s multiple-comparison test was used for all HEK293T and iPSC-derived motor neuron experiments, with the exception of signal-intensity linear-regression analysis. Two-way ANOVA with Dunnett’s multiple-comparison test was used for the ELISA experiments. A *p*-value less than 0.05 was considered significant.

## Figures and Tables

**Figure 1 ijms-23-16013-f001:**
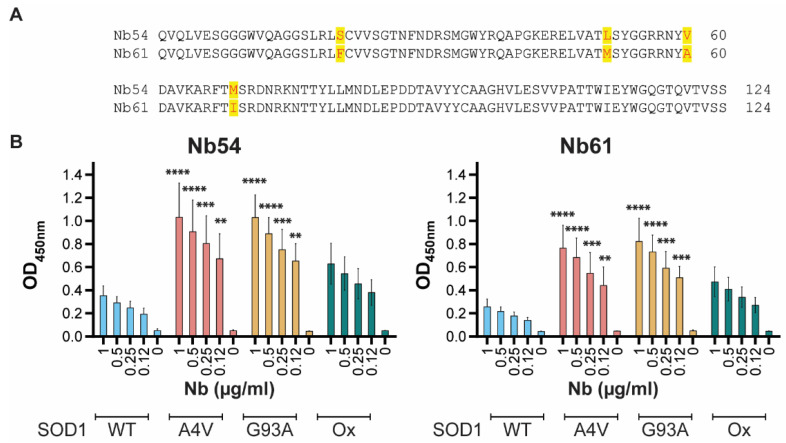
Nb54 and Nb61 are selective for ALS-linked SOD1 variants over SOD1 WT in vitro. (**A**) Amino-acid sequence alignment of Nb54 and Nb61. Sequence differences have red font and are highlighted in yellow. (**B**) Enzyme-linked immunosorbent assays (ELISAs) were performed by coating the indicated SOD1 protein onto the well and increasing the concentration of Nb54 (**left**) or Nb61 (**right**) as described in the methods. The optical density (OD) at 450 nm correlates with Nb reactivity for the indicated SOD1 variant. Bars depict mean ± standard deviation for three independent experiments. For each nanobody concentration (μg/mL), the OD at 450 nm for SOD1 A4V, G93A, or Ox was compared to the corresponding value for SOD1 WT using a two-way ANOVA followed by Dunnett’s multiple-comparison test. ** *p* < 0.01, *** *p* < 0.001, and **** *p* < 0.0001.

**Figure 2 ijms-23-16013-f002:**
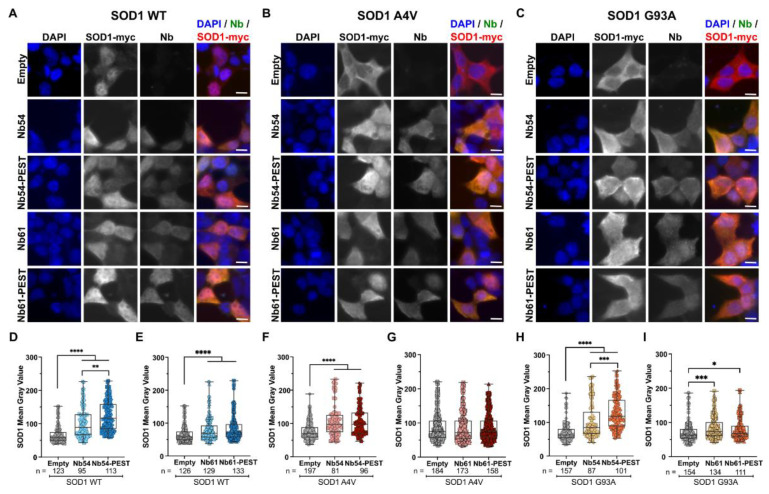
Exogenous SOD1 levels are elevated upon co-expression with anti-SOD1 nanobodies. (**A**–**C**) Immunofluorescence images of HEK293T cells co-transfected with myc-tagged SOD1 WT (**A**), SOD1 A4V (**B**), or SOD1 G93A (**C**) and nanobodies (Nb54, Nb54-PEST, Nb61, or Nb61-PEST). An “empty” nanobody vector condition serves as a control for baseline SOD1-myc expression. Scale bar, 10 μm. (**D**–**I**) Quantification of SOD1-myc fluorescence-signal intensity in cells co-transfected with nanobody and SOD1. (**D**) Co-transfection of Nb54 and Nb54-PEST leads to significantly enhanced SOD1 WT signal. (**E**) The same co-transfection as in (**D**) except with Nb61 and Nb61-PEST constructs. (**F**) Co-transfection of Nb54 and Nb54-PEST leads to a significantly enhanced SOD1 A4V signal. (**G**) Neither Nb61 or Nb61-PEST have an effect on SOD1-myc signal intensity when co-transfected with SOD1 A4V. (**H**) Co-transfection of Nb54 and Nb54-PEST leads to a significantly enhanced SOD1 G93A signal. (**I**) Same as (**H**) except with Nb61 constructs. For (**D**–**I)**, data are pooled from three biological replicates and represented with box-and-whisker plots, with boxes indicating the 25th (above) to 75th (below) percentiles and the median (line); whiskers denote the maximum and minimum values. Each point represents one cell (n refers to cell number beneath the plots); a different symbol is used for each biological replicate. Statistical analyses were performed with the Kruskal–Wallis test followed by Dunn’s to correct for multiple comparisons; * *p* < 0.05, ** *p* < 0.01, *** *p* < 0.001, and **** *p* < 0.0001. All significant comparisons are shown.

**Figure 3 ijms-23-16013-f003:**
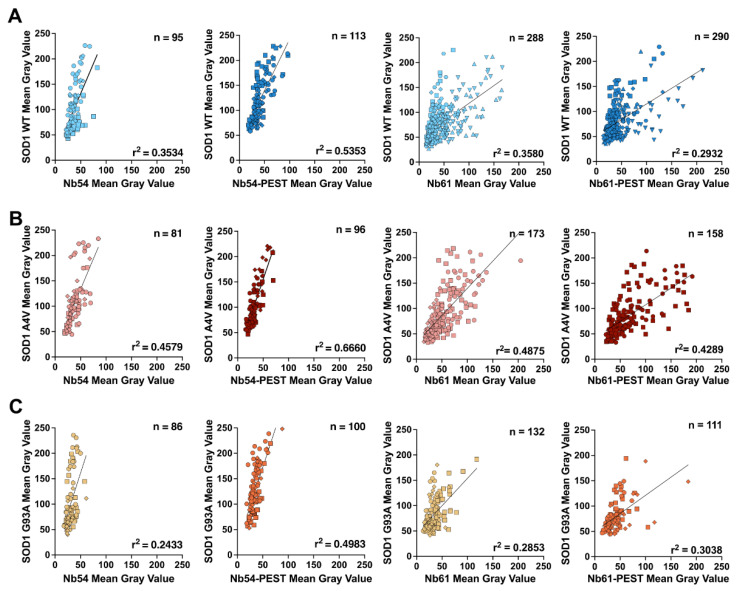
Exogenous expression levels of SOD1-myc and nanobody proteins are positively correlated in cellulo. Linear-regression analyses of SOD1 and nanobody signal intensity. For all conditions there is a moderate positive correlation, with SOD1 and Nb54-PEST co-transfections demonstrating the strongest correlations. (**A**) HEK293T cells co-transfected with SOD1 WT and Nb54, Nb54-PEST, Nb61, or Nb61-PEST. (**B**) HEK293T cells co-transfected with SOD1 A4V and Nb54, Nb54-PEST, Nb61, or Nb61-PEST. (**C**) HEK293T cells co-transfected with SOD1 G93A and Nb54, Nb54-PEST, Nb61, or Nb61-PEST. For (**A**–**C**): Data are pooled from three biological replicates with the exception of SOD1 WT co-transfected with Nb61 and Nb61-PEST, which are pooled from six biological replicates. Each point represents one cell; each symbol denotes a specific biological replicate. The r^2^ values reflect the fit of experimental data to the depicted regression line. Increasing r^2^ values indicate that the SOD1 and Nb signal intensities are positively correlated.

**Figure 4 ijms-23-16013-f004:**
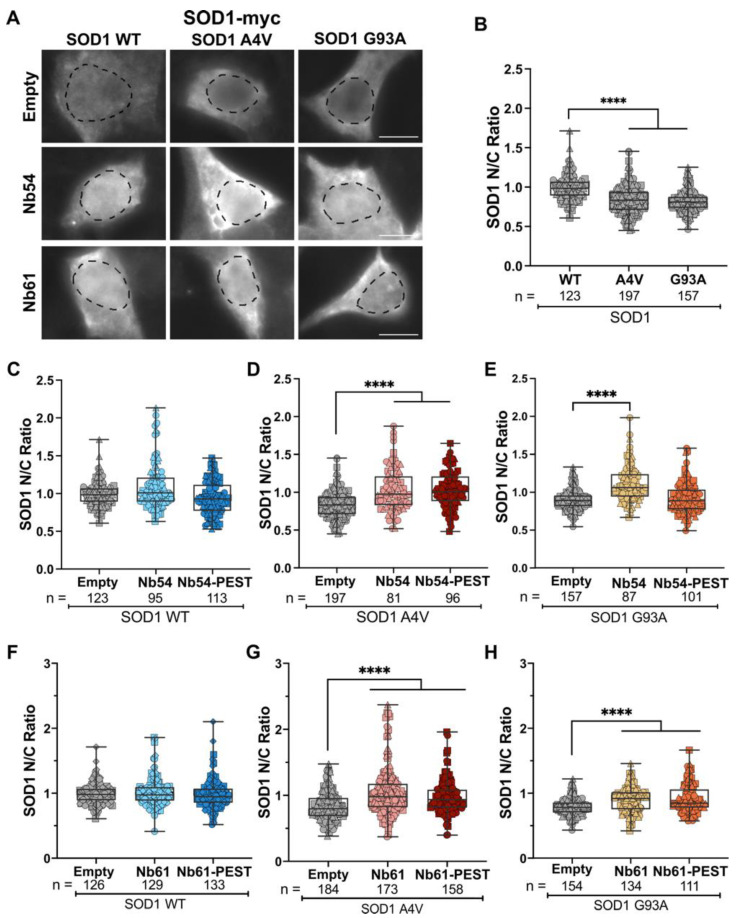
Nb54 and Nb61 restore the nucleocytoplasmic ratio of mutant SOD1 to SOD1 WT levels. (**A**) Immunofluorescence images of SOD1-myc signal in HEK293T cells co-transfected with SOD1 WT, SOD1 A4V, or SOD1 G93A and nanobodies (Nb54 or Nb61) or an empty vector control. Scale bar, 10 μm. (**B**) Quantification of SOD1-myc N/C ratio in cells co-transfected with SOD1 WT, SOD1 A4V, or SOD1 G93A and an empty vector control. SOD1 A4V and G93A have significantly reduced N/C ratios compared to SOD1 WT, indicating more cytoplasmic SOD1-myc signal. (**C**–**E**) Quantification of SOD1-myc N/C ratio in cells co-transfected with Nb54 or Nb54-PEST and SOD1 WT (**C**), SOD1 A4V (**D**), or SOD1 G93A (**E**). Co-transfection with Nb54 or Nb54-PEST increased the N/C ratios of both SOD1 variants, resulting in ratios similar to SOD1 WT, with the exception of Nb54-PEST with SOD1 G93A (**E**). (**F**–**H**) Quantification of SOD1-myc N/C ratio in cells co-transfected with Nb61 or Nb61-PEST and SOD1 WT (**F**), SOD1 A4V (**G**), or SOD1 G93A (**H**). Both Nb54 and Nb54-PEST significantly increased SOD1 A4V and SOD1 G93A N/C ratios to ratio values similar to SOD1 WT. For (**B**–**H**), data are pooled from three biological replicates and represented with box-and-whisker plots, with boxes indicating the 25th (above) to 75th (below) percentiles and the median (line); whiskers denote the maximum and minimum values. Each point represents one cell (*n* refers to cell number beneath the plots); a different symbol is used for each biological replicate. Statistical analyses were performed with the Kruskal–Wallis test followed by Dunn’s test to correct for multiple comparisons; **** *p* < 0.0001. All significant comparisons are shown.

**Figure 5 ijms-23-16013-f005:**
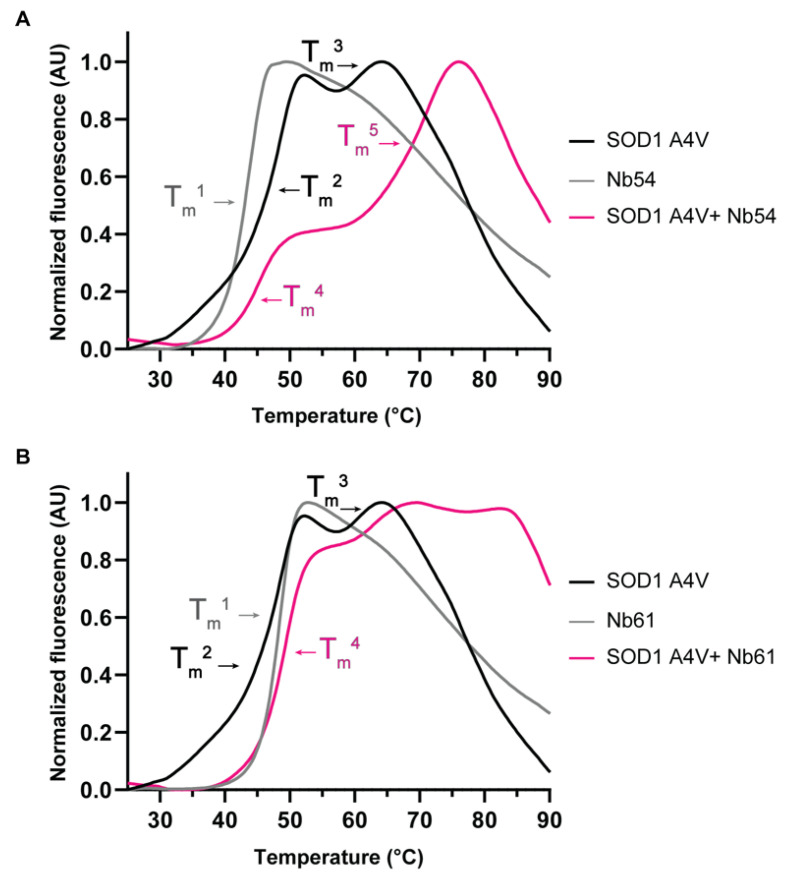
Binding of Nb54 and Nb61 confer stabilization to SOD1 A4V. Thermal denaturation profiles obtained by differential scanning fluorimetry (DSF) for SOD1 A4V and anti-SOD1 nanobodies alone and in complex. (**A**) SOD1 A4V (black curve), Nb54 (grey curve), and a mixture of SOD1 A4V and Nb54 (magenta curve). Addition of Nb54 to SOD1 A4V results in a higher-temperature (T_m_^5^) melting transition. T_m_^1^ = 43.8 °C, T_m_^2^ = 48.9 °C, T_m_^3^ = 60.0 °C, T_m_^4^ = 45.0 °C, T_m_^5^ = 71.1 °C. (**B**) As in (**A**) except with Nb61 (grey curve). Melting profile of Nb61/SOD1 A4V complex shows a peak at a higher temperature (~85 °C) compared to either SOD1 A4V or Nb61. T_m_^1^ = 48.6 °C, T_m_^2^ = 48.9 °C, T_m_^3^ = 60.0 °C, T_m_^4^ = 49.5 °C. Data presented here are representative of *n* = 3 separate experiments.

**Figure 6 ijms-23-16013-f006:**
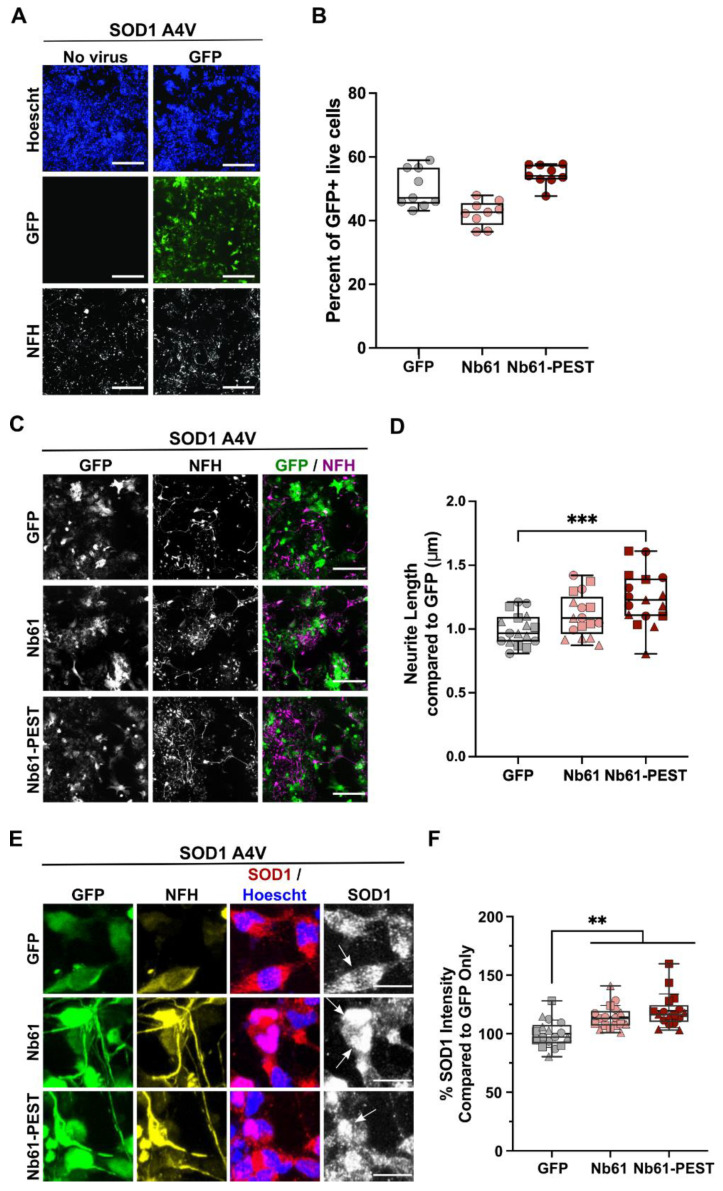
Nb61-PEST promotes neurite outgrowth in human SOD1 A4V motor neurons. SOD1 A4V iPSC-derived human motor neurons were thawed, plated into 384-well dishes, and assessed under various conditions after 7 days in culture. (**A**) Immunofluorescence images of human SOD1 A4V neurons that were untreated (**left**) or treated (**right**) with a control lentivirus expressing GFP. Neurofilament heavy (NFH, SMI32) staining identified motor neurons. Scale bar = 200 μm. (**B**) Quantification of the transduction efficiency for a control lentivirus expressing GFP or lentiviral constructs expressing either Nb61 or Nb61-PEST with a GFP reporter. Data are compiled from multiple wells (each point represents one well) from a representative biological replicate. (**C**) Immunofluorescence images of SOD1 A4V neurons that were transduced with the indicated lentivirus and stained as in (**A**). Scale bar = 200 μm. (**D**) Quantification of total neurite length for SOD1 A4V neurons transduced with the indicated lentivirus revealed significantly enhanced neurite outgrowth upon expression of Nb61-PEST compared to the GFP control virus. Data are compiled from *n* = 3 biological replicates; each replicate is denoted by a distinct symbol. (**E**) As in (**C**) with additional anti-SOD1 staining. Scale bar = 25 μm. Arrows indicate cells that are GFP+, NFH+, and have a clear SOD1 signal. (**F**) Quantification of endogenous SOD1 fluorescence signal intensity from images shown in (**E**) demonstrates that transduced Nb61 and Nb61-PEST lead to enhanced SOD1 expression in SOD1 A4V neurons. Data are pooled across *n* = 3 biological replicates, with each point representing data acquired within a single well. For D and F, statistical analyses were performed with the Kruskal–Wallis test and Dunn’s multiple comparison test; ** *p* < 0.01, *** *p* < 0.001.

**Figure 7 ijms-23-16013-f007:**
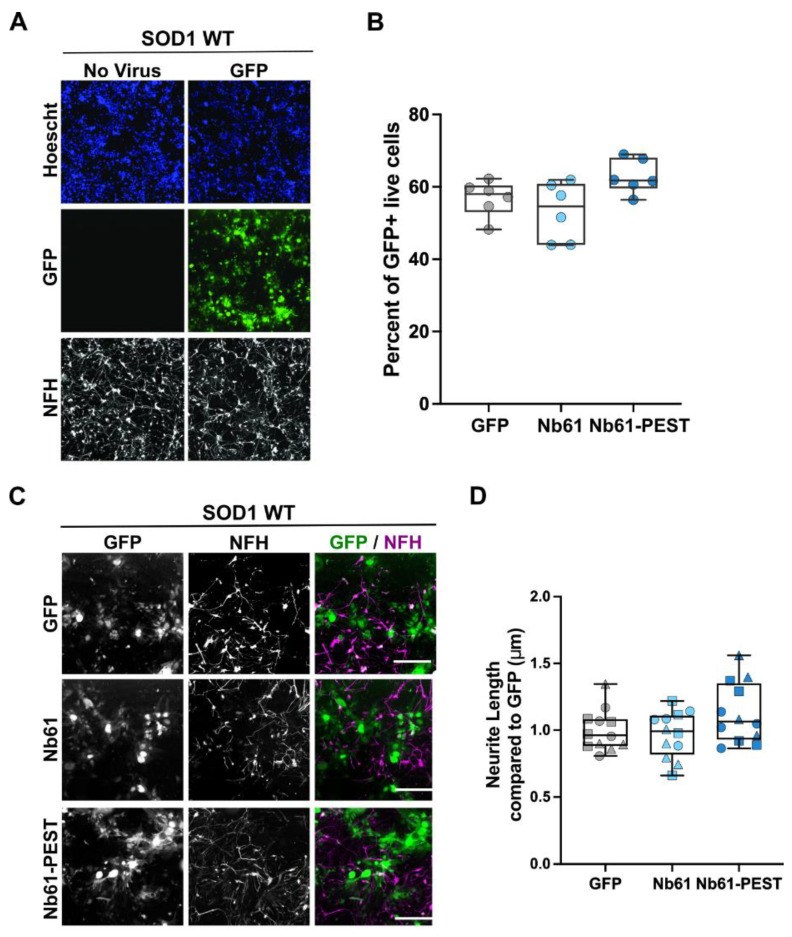
Nb61 constructs are nonlethal when expressed in human SOD1 WT motor neurons. SOD1 WT iPSC-derived human motor neurons were thawed, plated into 384-well dishes, and assessed as described for the SOD1 A4V neurons in Figure 6. (**A**) Immunofluorescence images of SOD1 WT neurons, either non-transduced (**left**) or transduced (**right**) with a control lentivirus expressing GFP. Motor neurons were identified with anti-NFH staining. Scale bar = 200 μm. (**B**) Quantification of the transduction efficiency for the indicated lentivirus from a representative biological replicate. (**C**) Immunofluorescence images of SOD1 WT iPSC-derived motor-neuron cultures at DIV7 that were transduced with a control GFP-expressing lentivirus or lentivirus co-expressing Nb61 or Nb61-PEST and GFP. NFH staining was used to identify motor neurons. Scale bar = 200 μm. (**D**) Quantification of total neurite length of SOD1 WT (*n* = 2) iPSC-derived motor neurons that were transduced with a control lentivirus expressing GFP or lentivirus co-expressing Nb61 or Nb61-PEST and GFP. Neither Nb61 nor Nb61-PEST altered neurite length compared to SOD1 WT neurons transduced with control GFP lentivirus. Data represent three experimental replicates, each denoted by a different symbol. Kruskal–Wallis with Dunn’s multiple comparison test was used for analysis; no significant comparisons were found.

**Figure 8 ijms-23-16013-f008:**
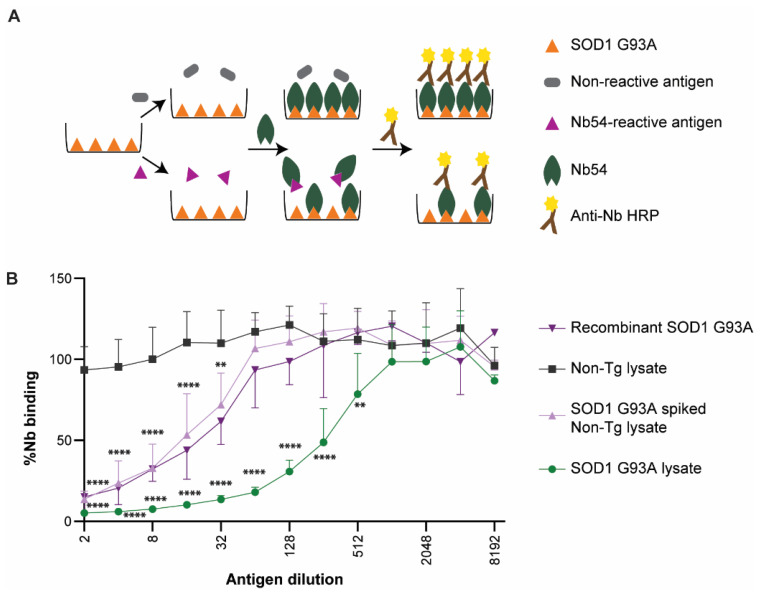
Nb54 detects SOD1 from spinal cord lysate of transgenic SOD1 G93A mouse. (**A**) A schematic representation of the competitive ELISA that was used to assess the competition of various antigens with SOD1 G93A for binding to Nb54. An antigen that is reactive to Nb54 is expected to reduce binding of Nb54 to SOD1 G93A, which is immobilized on the plate. (**B**) Spinal cord lysate from *SOD1^G93A^* transgenic mice (green line) expressing both human SOD1 G93A and endogenous murine SOD1 WT competes with (i.e., reduces) Nb54 binding to immobilized SOD1 G93A in a dose-dependent manner, whereas lysate from non-transgenic (Non-Tg; dark grey) mice expressing only endogenous murine SOD1 WT does not. Dilutions of Non-Tg lysate (light purple) or buffer spiked with recombinant SOD1 G93A (0.2 μg/25 μL; dark purple) were used as positive controls for competition of Nb54 binding to immobilized SOD1 G93A. Error bars depict standard deviation among *n* = 4 animals per genotype. A two-way ANOVA was performed for all lysate samples followed by a Dunnett’s multiple comparison test to compare *SOD1^G93A^* transgenic lysate or Non-Tg spiked with SOD1 G93A to the Non-Tg mouse lysate. ** *p* < 0.01, and **** *p* < 0.0001.

## Data Availability

All of the data referred to in this article are located within the main text. If there are requests for additional unpublished data not referred to in this article, please contact the corresponding author at daryl.bosco@umassmed.edu.

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
