# Peer review of "Anti-SOD1 Nanobodies That Stabilize Misfolded SOD1 Proteins Also Promote Neurite Outgrowth in Mutant SOD1 Human Neurons"

_ijms, 2022, doi:10.3390/ijms232416013_

Round 1
Reviewer 1 Report (Previous Reviewer 1)
Please ensure that the statistical analysis is included in the figures and not in the figure legends.
Author Response
This has been addressed in the updated manuscript file. Thank you for your review of this manuscript, we appreciate your time and comments.
Reviewer 2 Report (Previous Reviewer 2)
Authors provide a comprehensive analysis of different nanobodies against SOD1 both in vitro and in vivo, providing evidence for the potential of this anti-SOD1 proteins for preclinical studies
Author Response
Thank you for your review of this manuscript, we appreciate your time and comments.
Reviewer 3 Report (Previous Reviewer 3)
In this paper, Kumar and colleagues developed new SOD1 nanobodies that are selective against mutant and misfolded SOD1, a protein involved in ALS, over wild-type SOD1. Considering that there is no cure for this insidious disorder, this could be a potential interesting approach, but in my opinion, there are some discrepancies in the results that need to be addressed:
1. Fig.1B: They state that both Nb54 and Nb61 have a two fold higher reactivity toward SOD1ox compared to SOD1 WT. It is a strong conclusion considering that there is not statistical significance. How can they explain this discrepancy?
2. For most of their experiments they use an immortalized cell line the HEK293T a non-neuronal cell line. I understand that these cells have high transfection efficiency, but all the results obtain with this cell line, in my opinion, do not have a translational outcome. These cells, in fact, have completely different characteristics than a motor neuron cell line. There are more suitable cell lines they could have used, like for example, the NSC34 motor neuronal-like cells that are closer in morphology and characteristics to human motor neurons.
3. Fig.2: They found an increase in exogenous SOD1 levels in the cells. Did they check if this accumulation was toxic for the cells? This is important because in the first paragraph of the discussion they stated that” using a different anti-SOD1 from another source” they saw no changes in SOD1 levels”. If this was the case, their nanobodies once injected in vivo could cause toxic reactions due to the accumulation of SOD1. Could the authors comment on this?
4. It is not clear why in some experiments they use both type of nanobodies NB54 and Nb61 and in others like Fig.7 and 8 they use only one of the two. The experimental design of the whole manuscript appears to be a little confusing.
Author Response
Thank you for your review of this manuscript. I have included the responses here. I have also uploaded the full history of Reviewer comments and my responses from both rounds of review.
Comment 1. Fig.1B: They state that both Nb54 and Nb61 have a two fold higher reactivity toward SOD1ox compared to SOD1 WT. It is a strong conclusion considering that there is not statistical significance. How can they explain this discrepancy?
Response: We agree that strong conclusions regarding fold-changes cannot be made in this case and we have updated the text, lines 129-131 as follows:
Both Nb54 and Nb61 tended to exhibit higher reactivity toward SODox compared to SOD1 WT, however, this difference in reactivity did not reach statistical significance.
Comment 2. For most of their experiments they use an immortalized cell line the HEK293T a non-neuronal cell line. I understand that these cells have high transfection efficiency, but all the results obtain with this cell line, in my opinion, do not have a translational outcome. These cells, in fact, have completely different characteristics than a motor neuron cell line. There are more suitable cell lines they could have used, like for example, the NSC34 motor neuronal-like cells that are closer in morphology and characteristics to human motor neurons.
Response: We understand the Reviewer’s comment and take their point. However, in our opinion, iPSC-derived human motor neurons are more disease-relevant and state-of-the-art than NSC-34 cells and therefore we opted to focus our efforts on human neurons instead of NSC-34 cells. It is true that early on in the project we could have opted for NSC-34 cells instead of HEK cells. However, NSC-34 cells have their own limitations, including that they are mouse (not human) and immortalized (not neurons). Indeed, the ALS field has moved away from using NSC-34 cells in recent years. My lab has published with NCS-34 cells before (Ward, Cell Death and Disease, 2014; Baron, Molecular Neurodegeneration, 2013) and the issues we encountered included a rounded morphology that made fluorescence imaging difficult and low transfection efficiency. Further, in the 2013 publication, we performed similar assays on both HEK and NSC-34 cells with similar and consistent outcomes. Therefore, in our view, the HEK cells were suitable for the current nanobody study, especially since we included human neurons in this study as well.
We also tried to justify our use of HEK cells in the last round of reviews as follows:
Comment 1.2 (from previous round of reviews) What is the rationale for performing the studies on HEK293T cells and not doing all of them on cells involved in ALS (motorneurons, astrocytes, etc).
Response (from previous round of reviews): The neuronal cultures require significant expertise and resources. Our strategy was to initiate experiments in HEK293T cells, which provided useful and robust data on the SOD1/Nb interaction in cellulo. To ensure our study was relevant to ALS, we also performed experiments in human iPSC-derived motor neurons expressing the common ALS-linked SOD1 A4V mutation (Figure 6) and with lysates derived from an ALS mouse model (Figure 8). We revised the text in the Results section as indicated below:
Results (lines 145-146): We initiated characterization of the SOD1/Nb interaction in cellulo using HEK293T cells, as this represents a tractable cell line with high transfection efficiency.
Comment 3. Fig.2: They found an increase in exogenous SOD1 levels in the cells. Did they check if this accumulation was toxic for the cells? This is important because in the first paragraph of the discussion they stated that” using a different anti-SOD1 from another source” they saw no changes in SOD1 levels”. If this was the case, their nanobodies once injected in vivo could cause toxic reactions due to the accumulation of SOD1. Could the authors comment on this?
Response: We tried to address potential adverse consequences or toxicity of anti-SOD1 antibodies in the previous round of reviews as listed below. Based on the results in cell culture, the nanobodies do not exhibit toxicity. The effects of the nanobodies in vivo would need to be tested in a future preclinical study.
Comment 1.3 (from previous round of reviews) Is it good to stabilize the mutant and misfolding forms of SOD-1? As failure to send them to degradation, accumulation of these proteins could affect cell function and make them also non-functional.
Response (from previous round of reviews): We understand the Reviewer’s point. However, our data suggest that the nanobodies are stabilizing the mutant conformation in a manner that causes the mutant protein to adopt a more wild-type-like conformation. This is supported by our finding that the nuclear-to-cytoplasmic (N/C) localization of mutant SOD1 is similar to that of SOD1 WT when SOD1 is co-expressed in cells with the nanobodies. Indeed, others have demonstrated that enhanced cytoplasmic localization of mutant SOD1 correlates with pathological misfolding of the protein, as mentioned in our Discussion section. Further, the notion that stabilizing mutant SOD1 has therapeutic utility is supported by other studies in the literature, which are now referenced (#60-62) in the Discussion. Excerpts from the manuscript that address these points are pasted below.
Introduction section (lines 88-95): Anti-SOD1 nanobodies did not reduce expression levels of misfolded SOD1 protein in mammalian cells, but rather appear to stabilize and potentially mitigate the misfolded conformation of mutant SOD1 in cells and in vitro. Co-expression of anti-SOD1 nanobodies led to increased levels of mutant SOD1 in mammalian cells, consistent with enhanced SOD1 stability. Further, co-expression of anti-SOD1 nanobodies increased the nuclear-to-cytoplasmic (N/C) localization of mutant SOD1 to that of SOD1 WT, suggesting mutant SOD1 adopts a more wild-type conformation when in complex with the nanobodies.
Discussion (lines 420-424): Cytoplasmic localization of mutant SOD1 is likely a result of mutation-induced misfolding, which could expose a putative nuclear export signal and thus nuclear export of mutant SOD1 via CRM1 (Chromosomal Maintenance 1) [23]. Co-expression of our anti-SOD1 nanobodies restored mutant SOD1 in the nucleus to SOD1 WT levels.
Cont., (lines 429-431): …we speculate that binding of Nb54/61 to mutant SOD1 converts misfolded SOD1 into a more SOD1 WT-like conformation, thereby favoring the nuclear localization observed for SOD1 WT.
Discussion (lines 449-452): … a beneficial effect has been observed in ALS-SOD1 models upon treatment with diacetyl-bis(4-methylthiosemicarbazonato) copper(II) [Cu(II)(atsm)], which promotes metalation and increases the levels of SOD1 [60-62].
Comment 1.4 (from previous round of reviews) Does the use of the nanobodies, have any effects on cell function and survival?
Response (from previous round of reviews): We performed experiments with human iPSC-derived motor neurons expressing the common ALS-linked SOD1 A4V mutation (Figure 6) and found that expression of Nb61 led to significantly longer/more neurites in these mutant cells. Neurite outgrowth and length is a measure of cell health often used in the literature to assess toxicity and/or viability (reference #53). We also know that expression of nanobodies is not toxic, as for both SOD1 A4V (Figure 6) and SOD1 WT (Figure 7) the expression of nanobody either improved (Figure 6) or had no effect (Figure 7) on neurites. These outcomes warrant future studies to further assess neuron survival over longer periods of time and neuronal survival in the SOD1 G91A mouse model. An excerpt from the manuscript that address this point is pasted below.
Results (lines 299-303): Seven days post viral transduction, we assessed neuronal health by comparing total neurite lengths (anti-NFH) between constructs (Figure 6C) [53]. Compared to SOD1 A4V neurons expressing the GFP control lentivirus, SOD1 A4V neurons expressing Nb61 and Nb61-PEST exhibited a greater total neurite length, which reached statistical significance with Nb61-PEST (Figure 6D).
Comment 4. It is not clear why in some experiments they use both type of nanobodies NB54 and Nb61 and in others like Fig.7 and 8 they use only one of the two. The experimental design of the whole manuscript appears to be a little confusing.
Response: We did not receive this comment in the previous round of reviews, or from other Reviewers, and therefore we are a little surprised by this comment. The two nanobodies we characterize here exhibit similar reactivities. In the analyses where both nanobodies are used, similar outcomes are observed. In these circumstances, it is not unusual to move forward with more sophisticated (in terms of time, expertise and resources) experiments such as those with human neurons using one biologic.

Round 2
Reviewer 3 Report (Previous Reviewer 3)
Even if I am not really satisfied with the use of HEK cells for such a study, I have to agree with the justifications the authors made for using this immortalized cell line to study nanobodies.
Author Response
Thank you for your fast response and understanding.
This manuscript is a resubmission of an earlier submission. The following is a list of the peer review reports and author responses from that submission.
Round 1
Reviewer 1 Report
The authors explore the utilization of anti-SOD1 nanobodies for the development of novel tools for ALS treatment. The idea is interesting for the area. However, there exist several concerns for this reviewer.
1- The results shown in figure 1 seem like Nb54 can recognize either G93A and oxidized SOD to similar extents thus decreasing the potential use of this nanobody in the following studies across the manuscript. The error bar at higher concentrations and similar OD at lower concentrations makes similar results in both cases. Also, why Nb61 was not evaluated for oxSOD? Also, n=2 must be increased. Statistical analysis is needed in Figure 1
2- What is the rationale for performing the studies on HEK293T cells and not doing all of them on cells involved in ALS (motorneurons, astrocytes, etc).
3- Is it good to stabilize the mutant and misfolding forms of SOD-1? As failure to send them to degradation, accumulation of these proteins could affect cell function and make them also non-functional.
4- Does the use of the nanobodies, have any effects on cell function and survival? The authors must evaluate these issues.
Author Response
Reviewer #1
The authors explore the utilization of anti-SOD1 nanobodies for the development of novel tools for ALS treatment. The idea is interesting for the area. However, there exist several concerns for this reviewer.
We sincerely thank the Reviewer for their time. We are also grateful for your comments that have improved the quality of this work.
Comment 1.1 The results shown in figure 1 seem like Nb54 can recognize either G93A and oxidized SOD to similar extents thus decreasing the potential use of this nanobody in the following studies across the manuscript. The error bar at higher concentrations and similar OD at lower concentrations makes similar results in both cases. Also, why Nb61 was not evaluated for oxSOD? Also, n=2 must be increased. Statistical analysis is needed in Figure 1
Response: We agree with the Reviewer that additional experiments should be included for rigor and statistical analysis. The ELISA result (revised Figure 1B) now includes data from three independent ELISA experiments. We also included SODox for analysis of nanobody (Nb) 61, in addition to Nb54. For all Nb concentrations above 0, the comparison between SOD1 A4V and WT, as well as the comparison between SOD1 G93A and WT, were statistically significant for both Nb54 and Nb61. Comparisons between SODox and SOD WT were not statistically significant. We updated the figure legend and the text to reflect these results. Please note: we did not include stars (*) or bars in the graph to denote statistical significance as it was too “busy” and hard to discern. Excerpts from the manuscript that address these points are pasted below.
Figure legend 1B: Enzyme linked immunosorbent assays (ELISAs) were performed by coating the indicated SOD1 protein onto the well and increasing the concentration of Nb54 (left) or Nb61 (right) as described in the Methods. The optical density (OD) at 450nm correlates with Nb reactivity for the indicated SOD1 variant. Bars depict mean ± standard deviation for three independent experiments. For each nanobody concentration (mg/ml), the OD at 450nm for SOD1 A4V, G93A or Ox was compared to the corresponding value for SOD1 WT using a two-way ANOVA followed by Dunnett’s multiple comparison test. SOD1 WT vs A4V and SOD1 WT vs G93A: p<0.0001 (for Nb54/61 concentrations 0.5-1 µg/ml), p<0.001 (for Nb54/61 concentration 0.25 µg/ml), p<0.01 (for Nb54/61 concentration 0.12 µg/ml). p>0.05 (not significant) for all other comparisons.
Results section (lines 128-136): Relative to SOD1 WT, both Nb54 and Nb61 exhibited 3-4 -fold higher reactivity toward SOD1 A4V and SOD1 G93A when tested at 0.12-1 µg/ml concentrations of the respective Nb (Figure 1B). Nb61 also reacted with the denatured form of both SOD1 WT and G93A (Figure S1). Both Nb54 and Nb61 exhibited an approximate 2-fold higher reactivity toward SODox compared to SOD1 WT, however this difference in reactivity did not reach statistical significance. Given that SOD1 A4V was not used as an immunogen for the generation of these nanobodies, the high reactivity of Nb54 and Nb61 for SOD1 A4V reinforces the notion that ALS-linked SOD1 variants share a common misfolded conformation [6,40].
Comment 1.2 What is the rationale for performing the studies on HEK293T cells and not doing all of them on cells involved in ALS (motorneurons, astrocytes, etc).
Response: The neuronal cultures require significant expertise and resources. Our strategy was to initiate experiments in HEK293T cells, which provided useful and robust data on the SOD1/Nb interaction in cellulo. To ensure our study was relevant to ALS, we also performed experiments in human iPSC-derived motor neurons expressing the common ALS-linked SOD1 A4V mutation (Figure 6) and with lysates derived from an ALS mouse model (Figure 8). We revised the text in the Results section as indicated below:
Results (lines 147-148): We initiated characterization of the SOD1/Nb interaction in cellulo using HEK293T cells, as this represents a tractable cell line with high transfection efficiency.
Comment 1.3 Is it good to stabilize the mutant and misfolding forms of SOD-1? As failure to send them to degradation, accumulation of these proteins could affect cell function and make them also non-functional.
Response: We understand the Reviewer’s point. However, our data suggest that the nanobodies are stabilizing the mutant conformation in a manner that causes the mutant protein to adopt a more wild-type-like conformation. This is supported by our finding that the nuclear-to-cytoplasmic (N/C) localization of mutant SOD1 is similar to that of SOD1 WT when SOD1 is co-expressed in cells with the nanobodies. Indeed, others have demonstrated that enhanced cytoplasmic localization of mutant SOD1 correlates with pathological misfolding of the protein, as mentioned in our Discussion section. Further, the notion that stabilizing mutant SOD1 has therapeutic utility is supported by other studies in the literature, which are now referenced (#60-62) in the Discussion. Excerpts from the manuscript that address these points are pasted below.
Introduction section (lines 88-95): Anti-SOD1 nanobodies did not reduce expression levels of misfolded SOD1 protein in mammalian cells, but rather appear to stabilize and potentially mitigate the misfolded conformation of mutant SOD1 in cells and in vitro. Co-expression of anti-SOD1 nanobodies led to increased levels of mutant SOD1 in mammalian cells, consistent with enhanced SOD1 stability. Further, co-expression of anti-SOD1 nanobodies increased the nuclear-to-cytoplasmic (N/C) localization of mutant SOD1 to that of SOD1 WT, suggesting mutant SOD1 adopts a more wild-type conformation when in complex with the nanobodies.
Discussion (lines 424-428): Cytoplasmic localization of mutant SOD1 is likely a result of mutation-induced misfolding, which could expose a putative nuclear export signal and thus nuclear export of mutant SOD1 via CRM1 (Chromosomal Maintenance 1) [23]. Co-expression of our anti-SOD1 nanobodies restored mutant SOD1 in the nucleus to SOD1 WT levels.
Cont., (lines 433-435): …we speculate that binding of Nb54/61 to mutant SOD1 converts misfolded SOD1 into a more SOD1 WT-like conformation, thereby favoring the nuclear localization observed for SOD1 WT.
Discussion (lines 453-456): … a beneficial effect has been observed in ALS-SOD1 models upon treatment with diacetyl-bis(4-methylthiosemicarbazonato) copper(II) [Cu(II)(atsm)], which promotes metalation and increases the levels of SOD1 [60-62].
Comment 1.4 Does the use of the nanobodies, have any effects on cell function and survival?
Response: We performed experiments with human iPSC-derived motor neurons expressing the common ALS-linked SOD1 A4V mutation (Figure 6) and found that expression of Nb61 led to significantly longer/more neurites in these mutant cells. Neurite outgrowth and length is a measure of cell health often used in the literature to assess toxicity and/or viability (reference #53). We also know that expression of nanobodies is not toxic, as for both SOD1 A4V (Figure 6) and SOD1 WT (Figure 7) the expression of nanobody either improved (Figure 6) or had no effect (Figure 7) on neurites. These outcomes warrant future studies to further assess neuron survival over longer periods of time and neuronal survival in the SOD1 G91A mouse model. An excerpt from the manuscript that address this point is pasted below.
Results (lines 301-305): Seven days post viral transduction, we assessed neuronal health by comparing total neurite lengths (anti-NFH) between constructs (Figure 6C) [53]. Compared to SOD1 A4V neurons expressing the GFP control lentivirus, SOD1 A4V neurons expressing Nb61 and Nb61-PEST exhibited a greater total neurite length, which reached statistical significance with Nb61-PEST (Figure 6D).
Reviewer 2 Report
The paper from Kumar and coll entitled “Anti-SOD1 Nanobodies That Stabilize Misfolded SOD1 Proteins Also Promote Neurite Outgrowth in Mutant SOD1 Human Neurons” is focused on the development and characterization of two nanobodies, with selectivity for mutant and misfolded forms of human SOD1, coded by the ALS-causing gene identified first.
The use nanododies is considered a principle a flexible and ductile strategy to control protein aggregation, in misfolding diseases including ALS and Authors provide preliminary interesting data that, in the opinion of this referee, should be supported by a biochemical analysis:
- In vitro characterization of Nb54 and Nb61 binding to mutant SOD1 such be supported by biochemical evidence, such as immunoblot showing binding to high molecular weight SOD1 and/or immunoprecipitation
-According to the data in figure 2 Authors conclude that “Anti-SOD1 nanobodies lead to enhanced, rather than reduced, levels of ectopic SOD1 in cellulo”. This conclusion is based only on immunostaing experiment and needs to be supported by biochemical analysis.
Author Response
Reviewer #2
We sincerely thank the Reviewer for their time. We are also grateful for your comments that have improved the quality of this work.
Comment 2.1 In vitro characterization of Nb54 and Nb61 binding to mutant SOD1 such be supported by biochemical evidence, such as immunoblot showing binding to high molecular weight SOD1 and/or immunoprecipitation
Response: In the initial submission, we included three separate biochemical analyses using recombinant/ purified proteins under native (i.e., non-denaturing) conditions to demonstrate an interaction between SOD1 and the nanobodies, including an ELISA (Figure 1), the differential scanning fluorimetry (DSF) assay (Figure 5), and the competition ELISA (Figure 8). All three approaches demonstrate an interaction between the nanobody and the SOD1 protein being tested.
In response to the Reviewer’s suggestion for a Western blot analysis, we included new data where we probed SOD1 WT and SOD1 G93A by a denaturing Western blot analysis using Nb61 as a primary antibody, as presented in new Supplemental Figure 1. These results demonstrate reactivity of Nb61 with denatured forms of both variants, which migrate at the expected molecular weight of the SOD1 monomer. A Western blot analysis with a pan SOD1 antibody is included in this figure as a positive control. An excerpt from the manuscript that address this point is pasted below.
As these are run under denaturing conditions, we do not necessarily expect to see a robust preference or selectivity of the nanobodies for mutant SOD1 over WT SOD1 as we do for assays run under native conditions (as discussed in detail in our prior publication; Bosco et al, Nature Neuroscience, 2010).
Results (line 131): Nb61 also reacted with the denatured form of both SOD1 WT and G93A (Figure S1).
Comment 2.2 According to the data in figure 2 Authors conclude that “Anti-SOD1 nanobodies lead to enhanced, rather than reduced, levels of ectopic SOD1 in cellulo”. This conclusion is based only on immunostaining experiment and needs to be supported by biochemical analysis.
Response: We performed Western blot analysis on n=5 independent co-transfections in an effort to analyze the bulk population of cells with a biochemical read-out. This data is now shown in the new Supplemental Figure 2 and is presented in the results section as stated below. The outcomes of this analysis are variable, likely reflecting the variation of transgene expression across a population of cells that have undergone transient co-transfection. Therefore, in our view, analysis of single cells by immunofluorescence analysis is more accurate, particularly for the linear regression analysis wherein the levels of SOD1-myc are correlated to the levels of nanobody; this analysis accounts for the variability of transgene expression across the population of cells by measuring the levels of these proteins, and how these levels correlate with each other, on a per cell basis.
Results (lines 184-195): We also examined the SOD1-myc levels by Western blot analysis of the cell lysates from the HEK co-transfection experiments (Figure S2A). In contrast to the per cell fluorescence intensity analyses (Figure 2), the outcomes of the lysate-based Western blot analysis were variable among experiments (Figure S2B), likely due to the variation of transgene expression across a population of cells that have undergone transient co-transfection. To examine this further, we performed a linear regression analysis of fluorescence intensity corresponding to anti-myc versus anti-Nb for cells co-expressing either myc-tagged SOD1 WT (Figure 3A), SOD1 A4V (Figure 3B) or SOD1 G93A (Figure 3C) with the various nanobody constructs. For all SOD1-myc and nanobody comparisons, including nanobody-PEST constructs, there was a positive correlation between SOD1-myc fluorescence intensity and nanobody fluorescence intensity on a per cell basis (Figure 3), indicating that the nanobodies generally enhance SOD1-myc expression.
Reviewer 3 Report
This is an interesting paper by Kumar et al that shows how anti-SOD1 nanobodies exhibited selectivity for human mutant SOD1 over endogenous murine SOD1, thus supporting the pre-clinical utility of anti-SOD1 nanobodies for testing in animal models of ALS. The paper is clear, and the experiments are novel and well designed. My only concern is the data from Fig.8. The authors use only one transgenic mouse and one non-Tg to state that anti-SOD1Nb54 detects SOD1 from spinal cord lysate of transgenic SOD1 G93A mouse. To have accurate and stronger results the authors should repeat this experiment using at least 3-4 animals/group. There are also several typos that need to be corrected before resubmission.
Author Response
We sincerely thank the Reviewer for their time. We are also grateful for your comments that have improved the quality of this work.
Comment 3.1 This is an interesting paper by Kumar et al that shows how anti-SOD1 nanobodies exhibited selectivity for human mutant SOD1 over endogenous murine SOD1, thus supporting the pre-clinical utility of anti-SOD1 nanobodies for testing in animal models of ALS. The paper is clear, and the experiments are novel and well designed. My only concern is the data from Fig.8. The authors use only one transgenic mouse and one non-Tg to state that anti-SOD1Nb54 detects SOD1 from spinal cord lysate of transgenic SOD1 G93A mouse. To have accurate and stronger results the authors should repeat this experiment using at least 3-4 animals/group.
Response: We agree with the Reviewer that additional mice should be included for rigor and statistical analysis. We have revised Figure 8B to include lysates from four SOD1 G93A mice and four non-Tg control mice. Please note: we did not include stars (*) or bars in the graph to denote statistical significance between comparisons as it was too “busy” and hard to discern. The comparisons were explicitly stated in the legend as follows:
Figure legend 8B. Spinal cord lysate from SOD1G93A transgenic mice (green line) expressing both human SOD1 G93A and endogenous murine SOD1 WT competes with (i.e., reduces) Nb54 binding to immobilized SOD1 G93A in a dose-dependent manner, whereas lysate from non-transgenic (Non-Tg; dark grey) mice expressing only endogenous murine SOD1 WT does not. Dilutions of Non-Tg lysate (light purple) or buffer spiked with recombinant SOD1 G93A (0.2µg/25µl; dark purple) were used as positive controls for competition of Nb54 binding to immobilized SOD1 G93A. Error bars depict standard deviation among n=4 animals per genotype. A two-way ANOVA was performed for all lysate samples followed by a Dunnett’s multiple comparison test to compare SOD1G93A transgenic lysate or Non-Tg spiked with SOD1 G93A to the Non-Tg mouse lysate: Non-Tg vs SOD1G93A transgenic lysate: p<0.0001 (dilutions from 2-256 -fold), p<0.01 (the 512-fold dilution). Non-Tg vs Non-Tg lysate spiked with SOD1 G93A: p<0.0001 (dilutions from 2-16 -fold), p<0.01 (the 32-fold lysate dilution). p>0.05 for all other comparisons.
Comment 3.2 There are also several typos that need to be corrected before resubmission.
Response: We have reviewed the document closely and made our best effort to address any typos.